# The Chester Step Test Is a Reproducible Tool to Assess Exercise Capacity and Exertional Desaturation in Post-COVID-19 Patients

**DOI:** 10.3390/healthcare11010051

**Published:** 2022-12-24

**Authors:** Renata Peroy-Badal, Ana Sevillano-Castaño, Rodrigo Núñez-Cortés, Pablo García-Fernández, Rodrigo Torres-Castro, Jordi Vilaró, Isabel Blanco, Elena Gimeno-Santos

**Affiliations:** 1Hospital Virgen de La Torre—Hospital Universitario Infanta Leonor, 28031 Madrid, Spain; 2Departamento de Radiología, Rehabilitación y Fisioterapia, Universidad Complutense de Madrid, 28040 Madrid, Spain; 3Department of Physical Therapy, Faculty of Medicine, University of Chile, Santiago 8380286, Chile; 4Physiotherapy in Motion Multispeciality Research Group (PTinMOTION), Department of Physiotherapy, University of Valencia, 46010 Valencia, Spain; 5International Physiotherapy Research Network (PhysioEvidence), 08025 Barcelona, Spain; 6Department of Pulmonary Medicine, Hospital Clínic, University of Barcelona, 08036 Barcelona, Spain; 7Institut d’Investigacions Biomèdiques August Pi i Sunyer (IDIBAPS), 08036 Barcelona, Spain; 8Blanquerna School of Health Sciences, Global Research on Wellbeing (GRoW), Universitat Ramon Llull, 08025 Barcelona, Spain; 9Biomedical Research Networking Center on Respiratory Diseases (CIBERES), 28029 Madrid, Spain; 10Barcelona Institute for Global Health (ISGlobal), 08036 Barcelona, Spain

**Keywords:** COVID-19, Chester step test, exercise capacity, rehabilitation

## Abstract

Many people recovering from an acute episode of coronavirus disease (COVID-19) experience prolonged symptoms. Exercise testing is a feasible and cost-effective option for assessing exercise tolerance, fatigue, and dyspnea related to effort. Being that the Chester step test (CST) is a progressive, submaximal test for predicting aerobic capacity, it could be a good option to explore. This study aimed to determine the reproducibility of CST for assessing exertional desaturation and exercise capacity in patients post-COVID-19 disease. A cross-sectional study was conducted on post-COVID-19 patients. Two attempts of the CST were performed. The intraclass correlation coefficient (ICC) was used to assess agreement between the two tests. Forty-two symptomatic post-COVID-19 patients were included, the mean age was 53.8 ± 10.3 years, and 52% were female. There was no significant difference between both tests (*p* = 0.896). Twenty-four percent of participants (10 cases) had a clinically significant decrease in SpO_2_ at the first assessment, compared to 30.1% (13 cases) at the second, with no significant difference. An ICC of 0.993 (95% CI: 0.987 to 0.996) was obtained for the total number of steps in the CST.

## 1. Introduction

Many people recovering from an acute episode of COVID-19 experience prolonged symptoms, especially fatigue, dyspnea, and low exercise tolerance [1]. Albeit the disease is primarily respiratory, it affects multiple systems, such as cardiovascular or neurological, leaving a broad spectrum of sequelae that affect COVID-19 survivors in the short, medium, and long-term [1,2,3,4]. Given the potentially detrimental long-term consequences of COVID-19 (e.g., impaired functional capacity), rehabilitation teams must conduct an appropriate assessment to plan early rehabilitation interventions [5,6].

Exercise testing is a feasible and cost-effective option for assessing exercise tolerance, fatigue, and dyspnea related to effort [7,8]. The six-minute walk test (6MWT) is the most commonly used test in different respiratory, metabolic, cardiological, or neurological diseases [7]. These tests can be performed in low-resource contexts and have widely demonstrated their usefulness in evaluating physical capacity [7]. However, to provide specific information, functional or exercise capacity, a test must be chosen according to the characteristics, the setting, and the physiological expected answer of each subject. For example, the 6MWT requires close supervision and specific technical requirements, such as a 30-m corridor that is often unavailable in hospitals or rehabilitation centers and less at home [7]. Thus, the Chester Step Test (CST) has the advantage of requiring small space, in comparison with the 6MWT, and portable equipment compared with tests using treadmills or cycle ergometers [9]. 

The CST was designed initially to estimate exercise capacity in healthy participants but has also been used in patients with respiratory diseases [10,11]. However, its use in patients recovered from COVID-19 is scarce [9]. The use of the CST has been scarcely described in post-COVID-19 patients. In a study carried out on 27 subjects with a post-COVID-19 condition the CST correlated significantly with the 6MWT [12]. Additionally, McNarry et al. used the CST to determine oxygen consumption in post-COVID-19 patients who underwent an inspiratory muscle-training protocol [13]. Another reason that could explain the little use of the CST is that they exist because there are many stair tests, and they are infrequently standardized and implemented in clinical practice [11].

Regarding the learning effect, in a previous study in 83 healthy subjects, the CST was stable between retests, suggesting no learning effect [14]. Being that CST is a progressive, submaximal test for predicting aerobic capacity [15], it could be a good option to explore. On the other hand, field tests are widely used in following-up and intervention programmes such as rehabilitation [16,17], and because of that and the need to find simple, reliable, and effective measures, it is essential to find alternatives to the 6MWT to assess post-COVID-19 patients. This study aimed to determine the reproducibility of CST for assessing exertional desaturation and exercise capacity in patients post-COVID-19 disease. The main motivation of this study was to determine if this test can be an alternative to assess physical capacity and exertional desaturation in different contexts where other field tests cannot be performed.

## 2. Materials and Methods

A cross-sectional study was conducted on post-COVID-19 participants admitted for a follow-up program at the Hospital Virgen de la Torre (Madrid, Spain) between March and May 2021. The inclusion criteria were as follows: (1) age ≥ 18 years; (2) diagnosis of COVID-19 by a positive polymerase chain reaction (PCR) assay for SARS-CoV2 in nasal and pharyngeal swab samples; (3) sequelae of COVID-19 disease and candidate for a pulmonary rehabilitation program (i.e., clinical case of post-COVID-19 condition as defined by the WHO-led Delphi consensus) [18]. Participants with any pre-existing musculoskeletal or neurological conditions limiting the ability to perform the CST and those who refused to participate were excluded. All participants gave informed consent after explaining the study objectives and procedures. The Ethics Committee of our institution approved all procedures, which were carried out following the recommendations of the “Strengthening the Reporting of Observational Studies in Epidemiology” (STROBE) guidelines [19]. The sample size was calculated using the software G*Power 3.1. Based on the ICC estimation data using the method of Walter et al. [20], the inputs were (i) acceptable reliability of p0 = 0.60; (ii) expected reliability of p1 = 0.80; (iii) power of 90%; (iv) α risk = 0.05. Assuming a loss of 5% of the sample size, it was established that at least 40 individuals would be needed.

During the initial visit, demographic characteristics, underlying comorbidities, and history of hospitalization were collected by interview. Two attempts of the CST were performed on the same day, with 30 min of resting time between both, on a 20-cm step and the auditory stimuli set the pace of the test using the following sequence: first stage: 15 steps/min; second stage: 20 steps/min; third stage: 25 steps/min; fourth stage: 30 steps/min; and fifth stage: 35 steps/min [21]. Each stage was two minutes long (total test of 10 min). A finger-pulse oximeter was used to record pulse oxygen saturation (SpO_2_) and heart rate (HR) before and at the end of the test. A desaturation level greater than or equal to 4 points from rest and post-test < 90% was considered clinically significant [22]. The modified Borg scale (0–10) was used to measure dyspnoea and fatigue immediately before and after each test [23].

All statistical analyses were performed using SPSS version 22.0 (IBM Corporation, Armonk, New York, NY, USA). The data distribution was assessed using the Shapiro–Wilk test and expressed as mean and standard deviation or median with percentile 25-percentile 75 (P25–P75) according to the distribution. Differences between each measurement were assessed using Student’s *t*-test, the Mann–Whitney U-test, or the Chi-square test for normally distributed, non-parametric, or categorical variables, respectively. The intraclass correlation coefficient (ICC) was used to assess agreement between the two tests. The significance level was set at *p* < 0.05.

## 3. Results

Forty-two symptomatic post-COVID-19 patients were included, the mean age was 53.8 ± 10.3 years, and 52% were female (Table 1). The median time between COVID-19 infection and CST was 5.8 ± 0.6 months. Twenty-nine patients had a history of hospitalization (15 (6–39) days), and 16.7% of patients required ICU admission.

In the first test, the median (P25-P75) of the total number of steps in the CST was 123 (68–177), while in the second test, it was 129 (74–195). There was no significant difference between both tests (*p* = 0.896). Twenty-four percent of participants (10 cases) had a clinically significant decrease in SpO_2_ at the first assessment (i.e., ≥4 points), compared to 30.1% (13 cases) at the second, with no significant difference (Chi-square = 0.538, *p* = 0.463). An ICC of 0.993 (95% CI: 0.987 to 0.996) was obtained for the total number of steps in the CST. The decrease in SpO_2_ was −2.0 (−4.0–−1.0) points and −2.0 (−6.0–−1.0) in the first and second assessment, respectively, with no significant difference (*p* = 0.435). We found no differences between dyspnea and fatigue of the lower extremity at the beginning or at the end of the test.

## 4. Discussion

This study demonstrated that the CST is a reproducible test with no differences between the first and second attempts. Both tests agree, and the difference was only 1.04 steps. Although clinical guidelines recommend performing two tests when using field tests due to the possible learning effect [7], our findings suggest that performing only one attempt would be sufficient to assess exercise capacity and exertional desaturation in recovered COVID-19 patients.

Overall, the CST is a demanding protocol, which reports the performance achieved during a progressive test, and, therefore, could guide clinicians and physiotherapists in prescribing exercise (e.g., resistance training) to improve their aerobic capacity, especially in those patients with persistent symptoms after COVID-19 pneumonia [24].

Our results are in line with similar studies performed in other chronic respiratory diseases. Recently, in 66 patients with interstitial lung disease (65 ± 13 years; 48.5% male; FVC 79 ± 19% pred; DLCO 49 ± 18% pred), an excellent intra-rater (ICC = 0.95; 95% CI: 0.91–0.97) and inter-rater reliability (ICC = 1.0; 95% CI: 0.99–1.0) were found for the CST [25]. A similar study performed on 32 patients with COPD (mean ± SD 70 ± 9 years, FEV_1_ 46 ± 15% of pred) also showed excellent relative reliability (ICC of 0.99; 95% CI: 0.97–0.99) [26]. This finding indicates that CST provides consistent results across different CRD and excellent agreement between healthcare professionals [24].

Although 6MWT is the most commonly test for assessing exercise capacity [7], is not always available. In addition, several factors make it impractical, such as the need for long corridor [7]. Thus, CST is an ideal, simple, and cost-effective alternative for assessing exercise capacity when space and time are limited or when qualified personnel or sophisticated equipment (e.g., treadmills or cycle ergometers) are unavailable. It is also noteworthy that it is even feasible to perform it in a home setting, adding value due to its ease in any environment. However, future research should compare the CST with other field tests (e.g., the 6MWT or the incremental shuttle walking test (ISWT)) to obtain more robust recommendations.

Although the results showed that this test is an alternative to assess physical capacity and desaturation to exertion, it should be considered that this is probably not the case for all populations. For example, if people have musculoskeletal problems, especially in the knees, it might not adequately reflect physical ability. On the other hand, if people are obese, the load on the lower extremities when going up the steps is greater and may affect adequate performance during the test.

The main limitation of this study was the lack of data regarding oxygen kinetics during the test; this information could not be recorded due to a lack of access to this equipment. However, the physiological response to the decrease of SpO_2_ was similar in both measurements, which provides relevant information on the ability of the CST to detect exertional desaturation. In addition, the rest time between two CST tests is unclear, so we followed the recommendations of clinical guidelines suggesting 30 min of rest between two tests in the 6MWT [7]. Therefore, as there was no significant difference between the two measurements, we can assume that the patients had already rested.

## 5. Conclusions

The CST is a reproducible test in patients recovered from an acute episode of COVID-19 who experience prolonged symptoms. Performing a single CST attempt may be sufficient to assess exercise capacity and exertional desaturation in this population. Therefore, CST may be used in the early stages of the evaluation of rehabilitation programs when there are economic or space constraints.

## Figures and Tables

**Table 1 healthcare-11-00051-t001:** Population characteristics and Chester step test in post COVID-19 patients (*n* = 42).

Characteristics.	Value	
**Age (years)**	53.8 ± 10.3	
Female sex (*n*,%)	20 (48%)	
BMI (Kg/m^2^)	29.3 (27.8–33.6)	
Hospitalized (*n*, %)	29 (70%)	
Hospital stay (days)	15 (6–39)	
ICU history (*n*, %)	7 (16.7%)	
**Comorbidities (*n*, %)** Hypertension Diabetes Cardiovascular disease Obesity	13 (31.0%) 9 (21.4%) 4 (9.5%) 22 (52.4%)	
**Chester Step Test: CST**	**First test**	**Second test**
Total steps Stages completed (0–5) SpO_2_ pre (%) SpO_2_ post (%) Change in SpO_2_ (points) Change in SpO_2_ ≥ 4, *n* (%) HR pre (beats/min) HR post (beats/min) Dyspnea pre (0–10) Dyspnea post (0–10) Fatigue pre (0–10) Fatigue post (0–10)	123 (68–177) 3.0 (2.0–4.0) 96 (96–97) 95 (92–97) −2.0 (−4.0–−1.0) 10 (23.8%) 83.7 ± 13.8 125 (104–144) 0 (0–2.0) 4.5 (3.0–7.0) 0 (0–2.0) 5 (3.0–7.0)	129 (74–195) 3.0 (2.0–4.0) 97 (96–98) 94 (90–96) −2.0 (−6.0–−1.0) 13 (30.1%) 84.7 ± 13.8 120 (104–142) 1.0 (0–2.0) 5.0 (3.0–5.0) 1.5 (0–3.0) 5.0 (3.0–5.0)

Abbreviations: BMI: Body mass index; CST: Chester step test; HR: Heart rate; ICU: Intensive care unit; SpO_2_: Oxygen saturation. Values are expressed as the mean ± SD or as the median (P25–P75) according to the distribution of the data.

## Data Availability

All relevant data related to this study have been provided in this article.

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
