# Peer review of "The Chester Step Test Is a Reproducible Tool to Assess Exercise Capacity and Exertional Desaturation in Post-COVID-19 Patients"

_healthcare, 2022, doi:10.3390/healthcare11010051_

Round 1

Reviewer 1 Report

The Chester step test is a reproducible tool to assess exercise capacity and exertional desaturation in post-COVID-19 patients

In introduction the authors can add the contributions and paper organization.

The authors can mention the motivations and challenges.

A finger pulse oximeter was used to record pulse oxygen saturation (SpO2) and heart rate (HR) before and at the end of the test. What are the ranges considered for the work.

The data distribution was assessed using the Shapiro-Wilk test and expressed as mean and standard deviation or median with interquartile range (IQR) according to the distribution. The authors can foot note these as an reference.

Twenty-four percent of participants (10 cases) had a clinically significant decrease in SpO2 at the first assessment (i.e.,≥4 points), compared to 30.1% (13 cases) at the second, with no significant difference (Chi-square = 0.538, p=0.463). The authors should explain the reason for the output achieved.

The main contribution of the work falls in results, the results should be improved by adding how they achieved better results in detail manner.

It would be better the authors can compare the work with similar works.

The authors can refer

 Deep learning and medical image processing for coronavirus (COVID-19) pandemic: A survey

An incentive based approach for COVID-19 planning using blockchain technology

Author Response

Response to reviewer 1

Reviewer 2 Report

The authors present the CST test as a cheaper and simpler alternative to other tests to assess the aerobic capacity of COVID patients. However, the limitations and not just the benefits of this test should be specified. The authors prove that it is enough to carry out only one trial, because the next one is no different from the first. Such a conclusion may be useful for clinicians and physiotherapists who do not have access to expensive equipment. The thesis was written in accordance with the requirements of scientific work and I have no formal objections,

58 - please explain how infrequently this test is used and, if so, why it is rarely used

60-61 - the goal is to check the reproducibility of the test in a group of COVID patients. This may suggest that there are already such ratings in other groups? Wouldn't it be better to design an experiment with a control group of healthy people and conclude that the test is reproducible, including in the group of COVID-19 patients? If there is no such data, it is worth highlighting it here

table 1 - it is not entirely clear what values are given, mean or median and what is in brackets?

The discussion lacks a comparison of test results with values obtained by other authors for a given age group. This will make the measurements more reliable.

134-136 Are there no results of such comparisons by other authors?

145-150 It is worth mentioning here the exclusions for the test. Are there any limitations to the use of this test for patients with severe disease, acute symptoms? Also people who are very overweight may have a problem with this test. This is why the walk test is often used because it does not put as much stress on the knees and hips as the CST test

Author Response

Response to reviewer 2

Round 2

Reviewer 1 Report

Paper can be accepted